# L-PEM: A Lightweight Model for Parametric Experiential Memory

## Abstract

LLMs excel across many tasks but typically lack the ability to accumulate and reuse prior experiences. As a result, they often reason from scratch, retracing known solution paths and repeating past mistakes. Existing work commonly relies on Retrieval-Augmented Generation (RAG) to retrieve experiential memory summarized by LLMs. However, this paradigm suffers from high latency and computational cost, utilizes memory based on relevance rather than utility, resulting in suboptimal outcomes. To address these issues, we propose **L-PEM** (A **L**ightweight model for **P**arametric **E**xperiential **M**emory), a novel approach that embeds experience into the parameters of a compact generative model. This architecture unifies memory generation and application in a single forward pass, effectively replacing the conventional store-and-retrieve paradigm. We train L-PEM with Group Relative Preference Optimization (GRPO) using rollouts from a frozen executor as feedback and evaluate it on multiple mathematical reasoning benchmarks. L-PEM delivers significant performance gains while maintaining low latency and computational cost. Extensive ablation and analysis further elucidate the mechanisms underlying L-PEM's effectiveness. [1]

## 1 Introduction

Large language models (LLMs) have achieved breakthrough progress across diverse tasks, including machine translation, information retrieval, and mathematical reasoning (Ahn et al., 2024; Naveed et al., 2025; Guo et al., 2025). Despite these advances, current LLMs lack mechanisms for accumulating and reusing task-specific experience. When faced with new problems, they tend to reason from scratch rather than leveraging prior experience, often re-exploring familiar solution paths and repeating similar mistakes (Feng et al., 2025; Fan et al., 2025; Yu et al., 2025). In contrast, humans possess a fundamental ability to distill procedural experience from past problems, internalize it, and apply it to new situations. This capacity allows us to avoid previous errors and respond more efficiently to novel challenges (Efklides, 2009; Kolb, 2014; Usher, 1993). Motivated by this, an ideal intelligent agent should similarly be able to accumulate and reuse experiential knowledge, enabling rapid adaptation to unfamiliar problems.

However, enabling models to effectively leverage prior experience remains a core challenge. This involves addressing two fundamental questions: how to extract meaningful experience from past interactions, and how to apply it effectively to new problems. A common approach in prior work is to prompt large models to retrospectively summarize their trajectories into explicit memories, which are then stored externally and retrieved at inference time using retrieval-augmented generation (RAG) techniques (Chhikara et al., 2025; Fang et al., 2025; Zhou et al., 2025). Despite their flexibility, these approaches face two key limitations: (i) Extracting experience via prompting LLMs incurs significant time and computational cost, which hampers scalability in complex settings (Wang et al., 2024a; Quinn et al., 2025); (ii) Retrieved experience is selected based on surface-level relevance, which can introduce noise or yield information that lacks practical utility (Du et al., 2025; Salama et al., 2025). Consequently, existing experience systems often deliver suboptimal outcomes.

To address these challenges, we propose **L-PEM** (A **L**ightweight Model for **P**arametric **E**xperiential **M**emory), a lightweight and modular mechanism for experiential learning. L-PEM is implemented

---

[1]We release out code at https://anonymous.4open.science/r/L-PEM

as a small, trainable generative language model that serves as an independent module. It internalizes domain-specific experience and generates personalized, task-conditioned guidance tailored to the capabilities of a frozen LLM executor. As Figure 1 illustrates, when a new task arrives, L-PEM leverages its internalized experience to generate a compact, structured memory prompt that guides the executor in solving the problem.

In contrast to previous methods, L-PEM embeds experiential memory implicitly in the parameters of the small model, rather than explicitly storing it as retrievable memory snippets. This design avoids the computational cost and potential summarization quality issues associated with prompting large models to retrospectively extract experience. Moreover, L-PEM generates task-specific guidance through a lightweight model and delivers it to the executor. This approach eliminates the reliance on relevance-based retrieval, enabling more precise and effective use of experiential memory. Crucially, only the parameters of the small model are updated during training, preserving the original capabilities of the LLMs and mitigating the risk of catastrophic forgetting. This modular setup also ensures stability, scalability, and ease of adaptation across domains.

To further improve L-PEM, we enhance Group Relative Preference Optimization (GRPO) (Guo et al., 2025). We use rollouts from the frozen executor to provide rewards for optimizing the memory module. For each problem, L-PEM generates multiple candidate experiential memories, the frozen executor produces an answer for each, and a simple reward function scores both answer correctness and memory schema completeness. We then transform the groupwise rewards into group-normalized advantages and perform a clipped policy-gradient update on the experiential memory policy. This shifts probability mass toward higher-quality experiential memory guidance while keeping the executor frozen.

We validate our approach on mathematical reasoning benchmarks, since mathematical reasoning is a domain where leveraging prior experience is particularly critical: solving complex problems often requires an accurate analysis of the problem, recalling effective solution patterns, avoiding previously made mistakes, and systematically applying heuristics accumulated from prior tasks. Compared with prior learning from experience methods, our approach consistently achieves state-of-the-art performance while remaining highly efficient. We further conduct extensive ablation and analysis experiments to probe the sources of the gains and to better understand the mechanisms underlying its effectiveness.

## 2 RELATED WORK

**Learning from Experience.** Learning from experience allows LLM-based agents to leverage prior knowledge on new tasks, enabling autonomous optimization, efficient reasoning, and stronger performance across diverse challenges; this paradigm is vital for building adaptive, intelligent agents in dynamic real-world settings (Tan et al., 2025; Zheng et al., 2023; Wang et al., 2024b; Liu et al., 2025). Previous studies have generally achieved learning from experience by directly training the agent itself, employing techniques such as reinforcement learning (Dong et al., 2025), imitation learning (Yang et al., 2024), and multi-agent learning (Fang et al., 2025). However, these training-based methods suffer from several limitations, including inadequate use of negative examples, poor cross-task generalization, catastrophic forgetting, etc. To address these issues, researchers have proposed learning-from-experience frameworks based on Retrieval-Augmented Generation (RAG) (Suzgun et al., 2025; Mishra et al., 2025; Chhikara et al., 2025). The pipeline can be summarized in three stages: (a) extraction, distilling past rollouts into structured experience snippets; (b) storage, placing these snippets in a dedicated memory bank as text or vector representations; and (c) retrieval and injection, fetching the most relevant snippets for a new task and injecting them into the prompt via in-context learning. However, several limitations persist: (i) the memory-management pipeline introduces substantial latency and compute overhead, constraining scalability in complex scenarios (Wang et al., 2024a; Quinn et al., 2025); (ii) experience is consumed by retrieving surface-level similar snippets rather than generating problem-conditioned content, often yielding vague or low-utility prompts (Du et al., 2025; Salama et al., 2025); and (iii) storage depends on hand-crafted templates that require continual revision as tasks evolve, which undermines generalization (Xu et al., 2025; Zhong et al., 2024).

**LLM and Reinforcement Learning.** The intersection of large language models (LLMs) and reinforcement learning (RL) is drawing increasing attention as researchers seek to move beyond static supervised fine-tuning and enable learning from dynamic, interactive feedback. RLHF (Ouyang et al., 2022) aligns LLM outputs with human preferences. Nowadays, a growing body of work adopts GRPO to drive learning under complex environment interactions and feedback, improving performance on long-horizon reasoning and tool use. Search-R1 (Jin et al., 2025) trains LLMs via RL to issue web search queries to maximize final-answer correctness. Tool-N1 (Zhang et al., 2025) formulates tool use as an RL problem so that the model learns when to call external tools or APIs. RAGEN (Wang et al., 2025) extends GRPO to multi-turn settings, strengthening multi-round self-evolution. Together, these studies show that RL moves LLMs from static imitation to interactive learning in complex environments, shaping sequential decision-making and experience utilization, and improving reasoning accuracy, tool competence, and self-improvement. Within learning from experience, RL is typically used in two ways: directly training the executor with RL (Lu et al., 2025), or training a stronger memory agent within a RAG framework to extract and inject memories (Yan et al., 2025). The limitations of both approaches have been discussed earlier. We propose L-PEM, which applies GRPO from a new perspective: it embeds experiential memory implicitly in the parameters of a lightweight model and directly generates problem-conditioned guidance for a frozen executor, improving both effectiveness and efficiency.

## 3 METHOD

### 3.1 PROBLEM FORMULATION

Once deployed, large language models (LLMs) operate in a fundamentally stateless manner: Consider a specific domain $D$, given a new problem instance $x \in D$, the executor LLM $E$ typically generates its output $y$ without conditioning on previously solved tasks or accumulated experience:

$$y \sim \pi_E(y \mid x).$$

This limitation arises because LLMs are bounded by a finite context window, which prevents them from retaining and reusing information across long interactions or evolving task sequences. Consequently, each new instance must be reasoned through from scratch, leading to redundant exploration of solution paths already discovered and frequent repetition of past errors.

This motivates us to equip LLMs with a memory system $M$ that saves and provides domain experience guidance to the frozen executor LLM. Formally, for an instance $x \in D$ and a specific LLM executor $E$, the memory system $M$ produces an auxiliary experiential memory prompt of limited length $m \sim \pi_M(m \mid x, D, E)$, and the executor $E$ then solves the task conditioned on this experience. The design of the memory system $M$ can thus be formulated as the following constrained optimization problem:

$$\max_{\pi_M} \mathbb{E}_{x \in D, \, m \sim \pi_M(\cdot|x,D,E), \, y \sim \pi_E(y|x,m)} \Big[ R\big(y, y^*(x)\big) \Big],$$

$$\text{s.t.} \quad \|m\| \leq L,$$

where $R(\cdot)$ denotes a reward measuring task correctness, and $L$ is a predefined maximum length of the generated experience.

Most existing work instantiates $\pi_M$ with the RAG pipeline: a LLM is first prompted to retrospectively summarize its trajectories into explicit memories, which are stored in an external bank. At inference time, the memory policy performs a search step that retrieves $k$ candidates from this store, followed by a generation step that rewrites the selected items into the prompt $m$. However, two key limitations remain: (i) High cost—prompting an LLM to mine experience from past traces is time- and compute-intensive, limiting scalability in complex settings; (ii) Shallow relevance—retrieval ranks items by surface similarity, often returning noisy or low-utility content that dilutes guidance.

### 3.2 ARCHITECTURE

We introduce **L-PEM**, a **L**ightweight Model for **P**arametric **E**xperiential **M**emory. Implemented as a small trainable generator, L-PEM functions as an independent module that internalizes domain

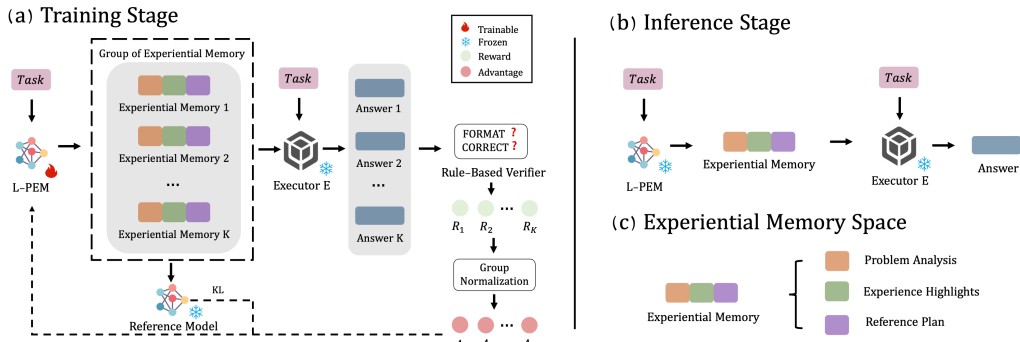

Figure 1: Illustration of L-PEM training and inference pipelines.

experience in its parameters and generates personalized, task-conditioned guidance for a frozen executor, shifting the responsibility of experience management from explicit retrieval systems into a compact generative module. As Figure 1 illustrates, when receiving a new input, L-PEM produces a compact, structured memory prompt that steers the executor's reasoning. Our design has three key characteristics: **(1) Implicit storage instead of explicit storage.** Unlike prior systems that explicitly distill experience into stored snippets, L-PEM retains experience implicitly in the lightweight model's parameters. This design avoids the computational costs of retrospective extraction by large models, the overhead of explicit storage, and the uncertain quality of summarization quality. **(2) Forward-pass experience generation instead of a RAG pipeline.** L-PEM uses a single forward pass to directly generate problem-solving guidance tailored to a specific executor, eliminating relevance-based retrieval in RAG pipelines; this yields more precise and effective guidance with lower system overhead and latency. **(3) Lightweight, pluggable modular adaptation.** We train L-PEM while keeping the executor $E$ frozen, which reduces training and deployment costs, enables hot-swapping across domains, and preserves $E$'s general competence and stability. Moreover, Operational rollouts can be readily logged and recycled as feedback, enabling continual training and steady evolution of L-PEM without modifying $E$.

**The Experiential Memory Space** As Figure 1 illustrates, L-PEM delivers experiential memory to the executor as a textual prompt. This design supports training and deployment even with a black-box executor and remains highly flexible, enabling plug-and-play integration without altering or destabilizing the executor's existing capabilities. Compared with RAG-style retrieved snippets, L-PEM's experiential memory is generated directly and adaptively tailored to the current problem, yielding greater specificity and precision. We structure the experiential memory space into three complementary components:

- *Problem analysis* — a concise assessment of the given instance $x$, highlighting its intrinsic difficulty, relevant substructures, and potential pitfalls;
- *Experience highlights* — domain-specific insights distilled from prior rollouts, specifying strategies, heuristics, or cautionary notes that can guide the executor $E$ when solving $x$;
- *Reference plan* — a concrete example of a step-by-step solution trajectory, illustrating how the problem can be decomposed and resolved.

This structured design ensures that L-PEM can generate guidance that is both high-level (analysis), prescriptive (experience highlights), and procedural (reference plan), thereby supporting the frozen executor $E$ in a comprehensive and task-conditioned manner.

### 3.3 TRAINING OF L-PEM

Our training objective for L-PEM is twofold: (i) extract experience from the executor E's rollouts in domain $D$ and encode this into L-PEM's parameters; and (ii) When a new task $x$ arrives, L-PEM leverages its parameterized experiential memory to generate a task-conditioned experiential memory prompt that conforms to our structured specification and is practically useful to executor $E$. We adopt Group Relative Preference Optimization (GRPO), using rollouts from the frozen executor as implicit supervision. As shown in Figure 1, for each problem, L-PEM proposes multiple candidate

experiential memory guidance; the executor solves conditioned on each. We design the reward to jointly account for the executor's answer correctness and the structural completeness of the memory produced by L-PEM. We then convert groupwise rewards into normalized, per-group advantages and apply a GRPO clipped policy update (with KL regularization) to shift probability mass toward higher-quality experiential guidance. Throughout, only L-PEM is updated while the executor remains frozen, distilling domain experience into a compact parametric form that provides on-demand guidance for new tasks.

**Reward Design.** The reward $R(y, m; x)$ is binary and reflects two criteria: (i) the executor $E$'s answer $y$ must be correct; (ii) the experiential memory $m$ produced by L-PEM must satisfy the structural requirements of the predefined memory space. The reward is defined as:

$$R(y, m; x) = \begin{cases} 1, & \text{if } y \text{ is correct and } m \text{ is structurally complete,} \\ 0, & \text{otherwise.} \end{cases}$$

**GRPO with group-normalized advantages.** For each training instance $x \in D$, we sample $K$ candidate experiential memories $\{m_j\}_{j=1}^{K} \sim \pi_{M,\theta}(\cdot \mid x, D, E)$. Each $m_j$ conditions the frozen executor to produce $y_j \sim \pi_E(\cdot \mid x, m_j)$, and we compute a scalar reward $R_j = R(y_j, m_j; x)$ that jointly scores answer correctness and memory schema completeness. We then convert groupwise rewards into per-group advantages via z-score normalization:

$$\bar{R} = \tfrac{1}{K} \sum_{j=1}^{K} R_j, \quad s = \sqrt{\tfrac{1}{K} \sum_{j=1}^{K} (R_j - \bar{R})^2} + \varepsilon, \quad A_j = \frac{R_j - \bar{R}}{s}.$$

**Objective with clipped policy update.** Let $\pi_{\text{old}}$ be the behavior policy for sampling $\{m_j\}$ and $\pi_{\text{ref}}(\cdot \mid x, D, E)$ a fixed reference policy (an copy of the base L-PEM). Define importance ratios $\rho_j = \frac{\pi_{M,\theta}(m_j \mid x, D, E)}{\pi_{\text{old}}(m_j \mid x, D, E)}$. The GRPO objective uses a PPO-style clipped update with KL regularization:

$$\mathcal{L}_{\text{GRPO}}(\theta) = -\frac{1}{K} \sum_{j=1}^{K} \min\Big( \rho_j A_j, \ \text{clip}(\rho_j, 1 - \epsilon, 1 + \epsilon) A_j \Big)$$

$$+ \beta \, \text{KL}\Big( \pi_{M,\theta}(\cdot \mid x, D, E) \, \big\| \, \pi_{\text{ref}}(\cdot \mid x, D, E) \Big).$$

where $\epsilon > 0$ is the clipping parameter, $\beta > 0$ controls the KL constraint.

**Training loop.** For each minibatch: (i) sample $K$ memories per $x$ from $\pi_{M,\theta}$; (ii) run the frozen executor to obtain $y_j$ and compute rewards $R_j$; (iii) form group-normalized advantages $A_j$ as above; and (iv) take one gradient step on $\mathcal{L}_{\text{GRPO}}(\theta)$. Throughout, only the L-PEM $M_\theta$ is updated while the executor $E$ remains frozen, shifting probability mass toward higher-quality memories and steadily distilling domain experience into a compact parametric form.

From a high-level perspective, we optimize one model using the observable execution behavior of another. With the executor $E$ frozen, its behavioral signals are used to train L-PEM, fostering tighter synergy between the two components. Concretely, we supervise an independent, lightweight generator with $E$'s problem-solving rollouts and correctness signals to produce high-quality experiential memory. We then apply GRPO over memory candidates (rather than answer candidates): groupwise rewards are normalized to form per-group advantages, and a clipped policy update increases the likelihood of better memories. This improves guidance and overall effectiveness without backpropagating through $E$. This decoupled paradigm enables L-PEM to continually improve from execution feedback without modifying the parameters of $E$, thereby preserving the executor's general capabilities while enhancing the system's end-to-end performance and robustness.

# 4 EXPERIMENT

## 4.1 EXPERIMENT SETTINGS

**Datasets.** We validate our approach in the domain of mathematical reasoning. Since mathematical reasoning is a domain in which prior experience is particularly consequential: solving complex prob-

lems demands precise problem analysis, recall of effective solution patterns, avoidance of previously observed failure modes, and the systematic application of heuristics accumulated from earlier tasks. For training, we randomly sample 5k instances from the DAPO dataset[2], which provides diverse problem statements with annotated solutions. Evaluation is conducted on four widely used mathematical reasoning benchmarks: GSM8K[3] (Cobbe et al., 2021), MATH[4] (Hendrycks et al., 2021), AIME24[5], AIME25[6]. These benchmarks jointly assess both routine arithmetic and high-level competition mathematics, allowing us to evaluate the generalization of every method across problems of varying difficulty, topics and solution structures. We use pass@1 as the primary metric, reporting the mean over 10 independent runs per dataset.

**Baselines.** In our main experiments, we compare L-PEM with several representative baselines. Together, these baselines span three families: (i) direct use of the original model without memory; (ii) direct train the executor, which can be viewed as enabling the executor to internalize and apply experience; and (iii) retrieval-augmented generation (RAG) approaches for learning from experience.

- **Original Model.** The frozen executor $E$ is directly applied to test problems without any memory augmentation or additional training. This serves as the minimal baseline.
- **Training-based method.**
  - **GRPO Original Model**: Directly train the executor $E$ using GRPO, baking prior experience into $E$'s parameters.
- **RAG-based methods.**
  - **MEM-0** (Chhikara et al., 2025): introduce a scalable long-term memory framework that dynamically extracts–integrates–retrieves salient information during dialogue to form a persistent external memory; this memory is represented as an entity–relation graph, enabling the capture of complex relations to support reasoning.
  - **Dynamic-Cheatsheet (DC)** (Suzgun et al., 2025): Augment black-box LMs with a persistent, self-curated test-time memory of concise, transferable strategies, enabling reuse without fine-tuning or labels. In our comparisons, we adopt the DC-RS (retrieve-and-synthesize) variant.
  - **Memento** (Mishra et al., 2025): Equip black-box LLMs with an external episodic memory, retrieving and adapting prior trajectories for case-based planning without fine-tuning. It supports learned Q-function retrieval with online write/read to guide tool-augmented execution.

**Implementaion Details.** We adopt QWEN3-1.7B (Yang et al., 2025) as the base models for L-PEM. In the main experiments, we use QWEN3-4B (Yang et al., 2025) and DEEPSEEK-R1-DISTILLED-LLAMA-3.1-8B (Guo et al., 2025) as downstream executors, which also serve as the backbone LLMs for the other baseline methods. We keep the thinking mode enabled for Qwen3 throughout all experiments. Unless otherwise specified, the maximum output length of the lightweight model is set to 2k tokens, while the maximum output length of the executor is set to 8k tokens. All experiments are conducted on 4 NVIDIA A100 (80GB) GPUs.

### 4.2 MAIN RESULTS

Table 1 shows across four math-reasoning benchmarks and two executors, L-PEM achieves the best performance on all datasets, demonstrating the effectiveness and generality across different executors. Relative to directly training the executor, L-PEM attains superior overall performance, indicating that internalizing the experience in a small parameterized generator that provides task-conditioned test-time guidance is effective and can even surpass directly optimizing the executor itself. Compared with RAG-based approaches, L-PEM delivers substantially stronger performance,

---

[2]https://huggingface.co/datasets/open-r1/DAPO-Math-17k-Processed

[3]https://huggingface.co/datasets/openai/gsm8k

[4]In this work, we use the subset MATH500. https://huggingface.co/datasets/HuggingFaceH4/MATH-500

[5]https://huggingface.co/datasets/HuggingFaceH4/aime_2024

[6]https://huggingface.co/datasets/math-ai/aime25

indicating that its mechanism of implicitly parameterizing experience and directly generating task-specific guidance addresses key limitations of RAG such as low-quality experience summarization and weak downstream guidance.

Regarding task difficulty, the largest performance gains of L-PEM emerge on the more challenging AIME problems, whereas on the comparatively saturated MATH500 and GSM8K benchmarks, L-PEM still provides modest improvements. This pattern suggests that experience-based guidance becomes increasingly critical and effective for the executor when tackling harder tasks. Collectively, these findings affirm the robustness and effectiveness of our design across diverse models and datasets within the math domain.

Table 1: Main results across four math-reasoning benchmarks, evaluated by pass@1. Scores are the mean of 10 independent runs per dataset.

| Methods | AIME24 | AIME25 | MATH500 | GSM8K |
|---|---|---|---|---|
| **QWEN3-4B** | | | | |
| Original Model | 0.407 | 0.380 | 0.784 | 0.932 |
| GRPO | 0.473 | 0.440 | 0.756 | 0.923 |
| MEM-0 | 0.413 | 0.403 | 0.792 | 0.938 |
| Dynamic-Cheatsheet | 0.447 | 0.417 | 0.803 | 0.943 |
| Memento | 0.463 | 0.433 | 0.811 | 0.947 |
| L-PEM - w/o - training | 0.423 | 0.387 | 0.801 | 0.942 |
| L-PEM | **0.517** | **0.467** | **0.822** | **0.951** |
| **DEEPSEEK-R1-DISTILLED-LLAMA-3.1-8B** | | | | |
| Original Model | 0.357 | 0.243 | 0.734 | 0.773 |
| GRPO | 0.437 | 0.323 | 0.795 | 0.852 |
| MEM-0 | 0.397 | 0.297 | 0.768 | 0.812 |
| Dynamic-Cheatsheet | 0.383 | 0.283 | 0.774 | 0.817 |
| Memento | 0.413 | 0.340 | 0.787 | 0.829 |
| L-PEM - w/o - training | 0.313 | 0.233 | 0.741 | 0.785 |
| L-PEM | **0.440** | **0.337** | **0.808** | **0.856** |

## 4.3 ABLATION STUDY

In the ablation study, we introduce an *L-PEM-w/o-training* baseline, which shares the same architecture as L-PEM but without training, to isolate the contribution of L-PEM training. Table 1 shows that this baseline behaves inconsistently across datasets and executors, yielding modest gains in some cases but clear regressions in others. The instability arises from its lack of domain experience; without training, the memory model often produces vague or erroneous strategies with overly coarse steps, and these shortcomings are especially damaging on harder tasks where a weaker executor cannot compensate. In contrast, the GRPO-trained L-PEM generates structured guidance that aligns with both the task and the executor, providing concise problem analyses, calibrated tactics, and step-wise plans. This targeted guidance avoids the pitfalls of the untrained model and consistently delivers significant improvements on challenging benchmarks.

## 5 ANALYSIS

To get deeper insight into L-PEM, we carry out a series of experiments that examine its efficiency, its compatibility with diverse model pairings, and its generalizability across models and domains.

## 5.1 EFFICIENCY

Table 2 summarizes efficiency along three axes: training efficiency, per-instance inference latency, and time-to-correct. During training, our method is substantially more efficient than directly fine-tuning a large model, because only the lightweight memory module is optimized while the executor

Table 2: Efficiency comparison on DEEPSEEK-R1-DISTILLED-LLAMA-3.1-8B with $4 \times$ A100 80GB, *measured on* AIME25. Training efficiency is reported as GPU-hours to convergence; per-instance time is the end-to-end latency for a single problem; time-to-correct is the average wall-clock time to obtain a correct answer.

| Method | Training Efficiency (GPU-hours) | Per-instance (s) | Time-to-correct (s) |
|---|---|---|---|
| Original Model | – | 0.84 | 3.46 |
| GRPO | 103 | 0.82 | **2.54** |
| MEM-0 | – | 3.80 | 12.84 |
| Dynamic-Cheatsheet | – | 4.10 | 14.91 |
| Memento | – | 3.90 | 13.13 |
| **L-PEM** | 52 | 1.01 | 3.00 |

Table 3: AIME24 pass@1 for pairing L-PEM scales with executors (Qwen3-4B-4k/8k, Qwen3-8B-8k). All lightweight models are trained independently with their respective paired executors.

| | Executor | | | Avg. |
|---|---|---|---|---|
| **Ligthweight Model** | **Qwen3-4B-4k** | **Qwen3-4B-8k** | **Qwen3-8B-8k** | **(row)** |
| **Qwen3-0.6B** | 0.323 | 0.479 | 0.643 | 0.478 |
| **Qwen3-1.7B** | 0.337 | 0.517 | 0.697 | 0.517 |
| **Avg. (col)** | 0.330 | 0.493 | 0.670 | 0.498 |

remains frozen, which reduces compute requirements. At inference time, L-PEM introduces a small, fixed, per-instance latency due to generating a structured memory that augments the executor. When measured by time-to-correct, L-PEM exhibits clear advantages: the auxiliary experience it provides raises task success rates and shortens average problem-solving time compared with RAG-based baselines. Overall, the design delivers notable gains in both training efficiency and solution speed while maintaining competitive inference latency.

## 5.2 MODEL PAIRING

We systematically examine the pairings of L-PEM's different lightweight models with various executors in Table 3[7]. On the side of smaller lightweight models, qwen3-0.6B yields some improvements, but its limited generalization and generation quality lead to experience memories that lack stability and coverage in complex tasks, resulting in only modest gains. In contrast, qwen3-1.7B exhibits stronger generalization and more structured generation, thereby providing more stable and significant improvements for the executor on harder tasks. On the side of executors, stronger LLMs are already capable of solving a substantial portion of problems correctly on their own, which reduces the marginal benefit brought by external experience, making the improvements from L-PEM less pronounced. Overall, there exists a complementary relationship between the scale of L-PEM and the capability of the executor: given a fixed executor, enhancing the generative capacity of L-PEM effectively amplifies the value of external experience; whereas when the executor itself is already sufficiently strong, the gains from external experience saturate, in which case further strengthening L-PEM (e.g., with a more powerful base model or improved training) may be necessary to achieve additional benefits.

## 5.3 GENERALIZABILITY

### 5.3.1 CROSS-MODEL

We further conducted a cross-executor transfer experiment to examine the portability of L-PEM across different executors. Specifically, we directly migrated the L-PEM trained in §4.2 with QWEN3-4B as the executor to another executors of different architecture or scale, without any additional training or adaptation. The results in Table 4 show that the transferred L-PEM can still provide a certain degree of performance improvement on the new executor, indicating that the generated experiential memory exhibits some level of generality. However, the improvement is relatively limited

---

[7]Qwen3-4B-4k denotes that the maximum output length is 4k tokens; the same convention applies hereafter.

Table 4: Cross-model transfer on AIME25: relative improvement (%) when applying L-PEM trained with QWEN3-4B as the executor to different executors. Relative improvement is computed as $\big[$(with L-PEM $-$ original)/original$\big] \times 100\%$.

| Dataset | DeepSeek-R1-Distilled-Llama-3.1-8B | GPT-4o | Claude-3.5-Sonnet |
|---------|-----------------------------------|--------|-------------------|
| AIME25  | +27.9                             | +22.7  | +25.3             |

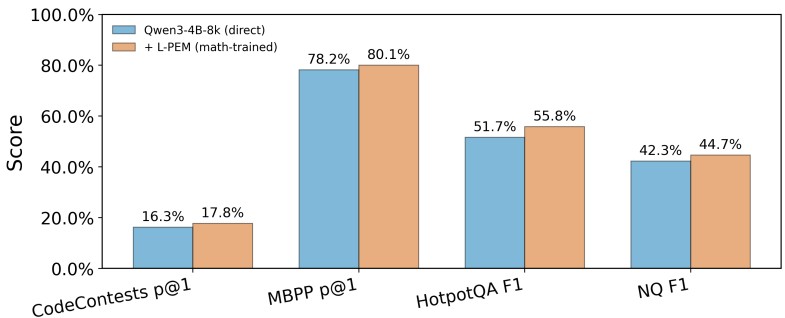

Figure 2: The cross-domain results for L-PEM with Qwen3-4B-8k as executor.

and remains below the performance of the original L-PEM–executor pairing (38.6%[8] vs. 27.9%). This limitation arises from L-PEM being optimized for a specific executor; variations in executors' capabilities and characteristics hinder portability and can degrade performance under direct transfer. To address this, we will explore joint training across multiple executors in future work.

### 5.3.2 CROSS-DOMAIN

We also investigated the cross-domain capability of L-PEM. We directly use the L-PEM trained in §4.2 with QWEN3-4B as the executor as well. As illustrated in Figure 2, although L-PEM is trained solely on mathematical data, it still enhances performance in two markedly different domains. This suggests that L-PEM acquires not just math-specific skills but a more general reasoning framework, including problem decomposition, explicit step planning, and the avoidance of common pitfalls. When the executor already possesses a baseline level of competence, these structured cues further improve its effectiveness. However, the cross-domain gains are relatively limited, because L-PEM lacks experiential memory for these unfamiliar domains. Enabling a single L-PEM to efficiently switch across and adapt to different domains is also a key direction for future work.

## 6 CONCLUSION

In this paper, we first highlight the limitations of current learning from experience methods: on the one hand, relying on large models to retrospectively summarize and extract experience incurs significant time and computational overhead, making it difficult to scale in complex settings; on the other hand, retrieval based on surface-level relevance often introduces noise or unhelpful information, leading to unstable experience reuse and limited gains. To address these issues, we introduce L-PEM, in which a small generative model internalizes experience into its parameters and produces structured, task-conditioned experiential memory to guide the executor. We train L-PEM with GRPO using rollouts generated by the frozen executor. Experiments on mathematical reasoning tasks demonstrate that L-PEM achieves both effectiveness and efficiency, while further ablation and analysis indicate its potential for extension to cross-model and cross-domain settings.

For future work, we plan to mainly focus on three directions: (i) validating our approach across a wider range of domains and executors, (ii) exploring mechanisms that enable a single L-PEM to achieve adaptive transfer across models and domains (e.g., parameter management) and (iii) designing robust continual-learning algorithms that allow L-PEM to evolve its experiential memory adaptively over time.

---

[8]The result is computed from the data reported in Table 1.

ETHICS STATEMENT

This work does not involve human subjects, user data collection, or interventions, and therefore does not require IRB approval. All datasets used are publicly available under their respective licenses; no personally identifiable information is processed. Our experiments focus on mathematical reasoning tasks and do not target sensitive attributes or demographics. We acknowledge that language models may reproduce or amplify biases present in training data; to mitigate this, we report datasets and evaluation settings, use publicly documented benchmarks, and release code for independent scrutiny and replication. The method does not enable surveillance, violate privacy, or intentionally facilitate harmful applications. We disclose compute resources in the appendix to support assessment of environmental impact. The authors are unaware of any conflicts of interest related to this work and adhere to the ICLR Code of Ethics.

REPRODUCIBILITY STATEMENT

We facilitate replication through three components. (i) Anonymized source code is linked in the abstract, enabling end-to-end execution. Training and evaluation can be reproduced by following the step-by-step instructions in the repository README. (ii) All datasets used in our experiments are linked in the §4 section, allowing direct download and the models employed are specified in the same section.

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

# APPENDIX

## A  DETAILED PROMPTS

In Figure 3 and Figure 4, we present the input prompts for L-PEM and the executor. Note that, for simplicity, we treat L-PEM's <think> section as the problem-analysis component of our memory space in practice.

## B  CASE STUDY

To examine the behavior of our method on real problem instances, we selected a geometry problem from Math500 as a case study. The problem is: in triangle $ABC$, with $AB = 17$, $AC = 8$, and $BC = 15$, let $D$ be the foot of the altitude from $C$ to $AB$. Find the area of $\triangle ACD$. The correct answer is $\frac{3840}{289}$.

In this case, the `think` section of L-PEM provided a concise and correct problem analysis: it first verified that $\triangle ABC$ is right-angled at $C$ by the Pythagorean theorem, and then suggested placing the triangle in a coordinate system (e.g., $C = (0, 0)$, $A = (0, 8)$, $B = (15, 0)$), finding the coordinates of $D$, and finally computing the area of $\triangle ACD$ using the shoelace formula (or equivalently, base–height/2). The `experience` section listed multiple possible solution strategies (similar triangles, area decomposition, coordinate geometry, etc.) and explicitly recommended the coordinate method as the most direct, efficient, and reliable approach for this problem. The `example` section further provided the concrete steps of the coordinate method (deriving the equation of line $AB$, the perpendicular line through $C$, their intersection point $D$, and then applying the area formula), yielding a reusable "action checklist."

The downstream executor successfully solved the problem and obtained the correct result $\frac{3840}{289}$. Notably, the executor was not additionally fine-tuned, and thus exhibited a degree of autonomy: it

---

**Input prompt for L-PEM**

```
You are both the **Memory Manager and the living memory itself**.
Here is the information you receive:

<problem>
{problem}
</problem>

Using this information, create **one memory item** with two parts:
1.  **<experience>** – a bullet list of distilled tips designed for
the downstream large-language solver that will help it solve the
current problem :  <problem> {problem} </problem> and similar tasks
in the future.  Draw on your knowledge of that solver's strengths
and weaknesses to give algorithms, heuristics, and pitfalls that
will most help it on future attempts.
2.  **<example>** – 3 -- 8 concise, high-level steps you judge to
be the most effective solution path for that solver on the problem:
<problem> {problem} </problem>
---

## Output Format

<experience>
• ...first distilled insight
• ...second distilled insight
• ...etc.
</experience>

<example>
1. Step 1 -- ...
2. Step 2 -- ...
3. ...
</example>
```

Figure 3: Input prompt for L-PEM.

---

**Input prompt for the executor**

```
The above is a Q&A dialogue between a user and a problem-solving
guidance model.
Treat the output of the guidance model as advisory context to solve
the problem:  prefer using its techniques when they fit, but you
may use alternative correct methods if they are more efficient or
clearer.  If you diverge from the advisory context, briefly explain
why.  Be concise and accurate.
Present your reasoning and answer in the following format:
<think> Content of Thinking</think><answer>[Final result
only]</answer>
```

Figure 4: Input prompt for the executor.

initially attempted two alternative approaches (e.g., area ratios/similar triangles and direct algebraic decomposition), but after cross-checking intermediate results, it eventually followed L-PEM's suggestion, switched to the simpler coordinate method, and completed the exact computation. This illustrates two points: (i) L-PEM outputs provide a clear strategic prior that guides the executor toward convergence when multiple strategies compete, and (ii) even without specialized training, the executor may perform exploratory reasoning, but under the structured guidance of L-PEM, it ultimately converges to the most efficient and verifiable solution path.

This case study demonstrates that L-PEM's structured memory and task decomposition not only improve efficiency (by reducing unproductive exploration) but also enhance interpretability and reusability of reasoning steps. The executor's limited autonomous exploration does not hinder final correctness; rather, in combination with L-PEM, it forms a cooperative paradigm of "prior guidance + verification-based convergence." This division of labor allows L-PEM to focus on *experiential strategy generation*, while the executor specializes in *step execution and verification*. Without additional fine-tuning, the system can still produce stable and correct solutions on benchmark problems.

---

**Case Study**

**Original Problem**

Given triangle $ABC$ with $AB = 17$, $AC = 8$, $BC = 15$. Let $D$ be the foot of the altitude from $C$ to $AB$. Find the area of triangle $ACD$.

---

**L-PEM Output**

<think>
The problem asks for the area of triangle ACD. In triangle ABC, the side lengths are AB = 17, AC = 8, and BC = 15, and D is the foot of the altitude from C to AB.

First, I need to determine the type of triangle ABC. Check the Pythagorean relation: $8^2 + 15^2 = 64 + 225 = 289 = 17^2$, so triangle ABC is right-angled at C.

This lets me place the triangle on a coordinate plane for convenience. Set $C$ at the origin $(0,0)$, $A$ at $(0,8)$, and $B$ at $(15,0)$. Then draw the perpendicular from $C$ to $AB$ and find the coordinates of the foot $D$.

With the coordinates of $A$, $C$, and $D$ in hand, I can compute the area of triangle $ACD$ using the shoelace formula or the ``base × height / 2'' formula. This is more direct and efficient than using trigonometry or similar triangles for this problem.
</think>

<experience>
• Identify the triangle type: use the Pythagorean theorem to check if it is a right triangle. Given $AB = 17$, $AC = 8$, and $BC = 15$, verify $8^2 + 15^2 = 64 + 225 = 289 = 17^2$; therefore, triangle $ABC$ is right-angled at $C$.
• Use coordinate geometry: place $C$ at $(0,0)$, $A$ at $(0,8)$, and $B$ at $(15,0)$, then find the coordinates of $D$, the foot of the altitude from $C$ to $AB$.
• Use the triangle area formula: since $ABC$ is a right triangle, its area is $\frac{1}{2} \times AC \times BC = \frac{1}{2} \times 8 \times 15 = 60$.
• Find the area of triangle $ACD$ using coordinates: with $A$, $C$, and $D$, apply the shoelace formula or the ``base × height / 2'' formula.
• Similar triangles or trigonometry could also be used, but in this case, coordinate geometry is more efficient.
</experience>

<example>
1. Step 1 --- Use the Pythagorean theorem to determine whether triangle $ABC$ is right-angled.
2. Step 2 --- Place the triangle on a coordinate plane with $C$ at $(0,0)$, $A$ at $(0,8)$, and $B$ at $(15,0)$.
3. Step 3 --- Find the coordinates of $D$, the foot of the altitude from $C$ to $AB$.
4. Step 4 --- Using the coordinates of $A$, $C$, and $D$, compute the area of triangle $ACD$ with the shoelace formula.
</example>

---

**Executor Output**

<think>

We are asked to find the area of triangle $ACD$ inside triangle $ABC$, where $AB = 17$, $AC = 8$, $BC = 15$, and $D$ is the foot of the altitude from $C$ to $AB$.

First, the user mentioned that the triangle is a right triangle. Let me confirm that. Using the Pythagorean theorem: $AC^2 + BC^2$ should equal $AB^2$. $8^2$ is 64, $15^2$ is 225, adding them gives 289, which is $17^2$. So yes, triangle ABC is right-angled at C. That simplifies things because the area of ABC is $(AC \times BC)/2 = (8 \times 15)/2 = 60$.

Now, the altitude from C to AB. In a right triangle, the altitude from the right angle to the hypotenuse has a known length. The formula for the altitude (h) is (product of the legs)/hypotenuse. So $h = (8 \times 15)/17 = 120/17$. But wait, that's the length of the altitude from C to AB, which is CD.

But the question asks for the area of triangle ACD. Since D is the foot of the altitude, CD is the height of triangle ACD with base AD. So I need to find AD.

In a right triangle, the length of the segments into which the altitude divides the hypotenuse can be found using similar triangles. The formula for AD is $(AC^2)/AB$. So $AD = (8^2)/17 = 64/17$.

Then the area of ACD is $(AD \times CD)/2$. Plugging in the values: $(64/17 \times 120/17)/2 = (7680/289)/2 = 3840/289$. Let me check that calculation again.

Alternatively, since the area of ABC is 60, and D divides AB into AD and DB. The area of ACD would be proportional to $AD/AB$ times the area of ABC. So $60 \times (AD/AB) = 60 \times (64/17)/17 = 60 \times 64/(17^2) = 3840/289$. That's the same result.

So the area of triangle ACD is $3840/289$. Let me compute that as a decimal to check if it makes sense. 289 goes into 3840 about 13.3 times. But since the problem likely expects an exact fraction, $3840/289$ is the answer. Wait, but maybe there's a simpler way. It is suggested that the coordinate method is simpler, so let me give it a try. Use coordinates: set $C = (0,0)$, $A = (0,8)$, $B = (15,0)$. Line $AB$ passes through $(0,8)$ and $(15,0)$, so its slope is $-\frac{8}{15}$ and its equation is

$$y = -\frac{8}{15}x + 8.$$

The altitude from $C$ is perpendicular to $AB$, hence has slope $\frac{15}{8}$ and equation

$$y = \frac{15}{8}x.$$

Intersecting these gives

$$\frac{15}{8}x = -\frac{8}{15}x + 8 \implies 289x = 960 \implies x = \frac{960}{289}, \qquad y = \frac{15}{8}x = \frac{1800}{289}.$$

Thus

$$D = \left(\frac{960}{289}, \frac{1800}{289}\right).$$

Compute the area of $\triangle ACD$ using the determinant (shoelace) formula with $A(0,8)$, $C(0,0)$, $D\left(\frac{960}{289}, \frac{1800}{289}\right)$:

$$[ACD] = \frac{1}{2}\left|x_A(y_C - y_D) + x_C(y_D - y_A) + x_D(y_A - y_C)\right| = \frac{1}{2} \cdot \frac{960}{289} \cdot 8 = \frac{3840}{289}.$$

Therefore, the area is $\boxed{\dfrac{3840}{289}}$. </think>

<answer> $\dfrac{3840}{289}$ </answer>

