# OpenReview forum: "L-PEM: A Lightweight Model for Parametric Experiential Memory"
_ICLR.cc/2026/Conference — ICLR 2026 Conference Withdrawn Submission_

### Official Review · Reviewer_BkLK · 2025-10-29

**Soundness:** 2
**Presentation:** 2
**Contribution:** 2
**Rating:** 4
**Confidence:** 3

**Summary:**

The authors propose L-PEM, aimed at addressing the issue of current LLMs' inability to effectively leverage prior experience when handling new mathematical tasks. L-PEM internalizes experience into the parameters of a lightweight model, avoiding the traditional memory storage and retrieval process, and directly generates task-conditioned guidance. The model is trained using GRPO, with rewards based on the accuracy of the answers generated by the downstream executor model and the format of the memory. It demonstrates high efficiency and accuracy in tasks such as mathematical reasoning.

**Strengths:**

1. L-PEM internalizes prior mathematical reasoning experience into a small model, avoiding the need for retrieval and knowledge storage required by previous methods.

2. When using a combination of qwen-1.5b and qwen-4b, the performance exceeds that of directly training qwen-4b with GRPO, demonstrating higher training efficiency.

**Weaknesses:**

1. The paper reports that the combination of Qwen3-1.7B + LLaMA-3.1-8B-distilled achieves results close to those of directly applying GRPO to train LLaMA-3.1-8B-distilled. This raises the question of whether the small performance gap may stem from the relatively limited capability of LLaMA-3.1-8B-distilled itself, which might prevent it from fully leveraging the external memory provided by L-PEM. The authors are encouraged to provide further explanation or ablation evidence supporting this point.

2. The core idea of this paper is to train a small model to store problem-solving memories for mathematical reasoning tasks. However, judging from the cases and prompts, this “memory” seems to function more like a structured reasoning plan or problem-solving strategy rather than a genuine long-term experiential memory. The authors do not clearly distinguish this difference from the conventional “think step by step” reasoning process in the paper, nor do they provide experimental analysis to highlight the conceptual or behavioral differences between the two approaches.

3. The current experiments rely on relatively small downstream models (e.g., the 4B-scale executor). It would be valuable to investigate whether L-PEM can scale effectively to larger downstream models. For instance, configurations such as L-PEM with Qwen3-1.7B paired with a Qwen3-8B executor, or L-PEM with Qwen3-4B paired with a Qwen3-14B executor, could better demonstrate scalability and the potential of the proposed approach under stronger reasoning backbones.

**Questions:**

Does the Qwen3 model use the "think" mode throughout the paper？

---

### Official Review · Reviewer_TWui · 2025-10-30

**Soundness:** 2
**Presentation:** 3
**Contribution:** 2
**Rating:** 4
**Confidence:** 4

**Summary:**

This paper proposes L-PEM (Lightweight Model for Parametric Experiential Memory). It introduces a novel architecture that shifts from explicit memory storage to implicit parametric memory. L-PEM consists of two components:

1. A small, trainable generative model (the L-PEM module) that internalizes domain experience within its parameters.
2. A larger, frozen LLM (the executor) that performs the task.

L-PEM is trained using GRPO (the executor is frozen). At inference time, L-PEM takes the task as input and generates a structured guidance prompt (comprising problem analysis, experience highlights, and a reference plan) in a single forward pass. This prompt guides the frozen executor.

The approach is evaluated on mathematical reasoning benchmarks (GSM8K, MATH500, AIME24/25). L-PEM demonstrates superior performance compared to RAG-based memory systems (MEM-0, Dynamic-Cheatsheet, Memento) and also outperforms directly training the executor with GRPO, while offering substantial efficiency benefits. Moreover, the trained L-PEM module can still provide some gain when paired with different executors in test time, and in cross-domain settings.

**Strengths:**

- Effective Replacement for RAG-based Memory. L-PEM successfully addresses the identified limitations of RAG systems. By replacing the store-and-retrieve pipeline with a generated, parametric memory, it achieves lower latency (Table 2) and avoids the noise associated with relevance-based retrieval, leading to stronger downstream performance than comparable RAG methods.
- Insightful Analysis of Transferability. The analysis in Section 5.3 provides valuable insights. The cross-model experiments (Table 4) show that an L-PEM trained with Qwen3-4B provides substantial relative improvements (22-27%) when transferred zero-shot to different models, including powerful ones like GPT-4o and Claude-3.5-Sonnet. This suggests the learned guidance captures generalizable strategies, not just executor-specific quirks.

**Weaknesses:**

- Limited Novelty: the novelty seems to lie in the application of this technique to the two-model setup (training a prompt generator based on the execution of a frozen model) rather than in algorithmic advancement.
- Lack of Domain-Shift / Capacity Evaluations: an advantage of non-parametric RAG systems is they are arguably more robust against domain shifts, as the knowledge is just text, and can be read and manipulated by any agent that operates with text. Moreover, the model size of L-PEM models used in the experiments is quite small (1.7B). It is unclear whether capacity will be an issue under the parametric approach in diversified task settings, which is less a problem for non-parametric RAG approaches.
- Missing Joint Training Ablation: the paper lacks crucial experiments comparing proposed approach to joint training of both L-PEM and the executor (perhaps using parameter-efficient methods like LoRA on the executor). Understanding the performance ceiling if both components were optimized together is necessary to fully characterize the trade-offs of the decoupled approach.

**Questions:**

- How does the performance of L-PEM change as the diversity and volume of the training data increase? Is there evidence of the 1.7B model capacity saturating, and how does this compare to the storage requirements of the RAG systems used as baselines?
- Could the authors comment on the potential benefits of unfreezing the executor during training? Even if full joint training is too expensive, would lightweight fine-tuning (e.g., LoRA) on the executor, guided by L-PEM, offer further improvements, or would it destabilize the training?

---

### Official Review · Reviewer_VX3K · 2025-10-30

**Soundness:** 2
**Presentation:** 3
**Contribution:** 2
**Rating:** 2
**Confidence:** 4

**Summary:**

This paper proposes the use of a lightweight model (L-PEM) ~1.7B) in conjunction with a frozen executor model that, given a task at hand, generates structured experiential memory prompts including problem analysis, experience highlights, and reference plans to guide the executor's reasoning.
The model is trained with GRPO using rollouts from the frozen executor as feedback, where rewards are based on both answer correctness and memory schema completeness. The evaluation shows modest improvements with respect to RAG-based memory baselines and direct executor training on mathematical reasoning benchmarks (GSM8K, MATH500, AIME24/25).

**Strengths:**

- I like a lot the idea of using a surrogate model to generate experiential memory. This modular approach where a small model provides guidance to a frozen larger model has potential and I think the authors scratched the surface of what could be possible with such an approach

- The paper is nice to read, and the concepts are well-defined, in the right order. The progression from problem formulation to architecture to training methodology is clear and logical.

**Weaknesses:**

- Conceptual mismatch between the promised "experiential memory" and actual implementation: although the idea of experiential memory is interesting, what the paper ended up implementing is, in a way, a separate reasoning model that provides the "<think></think>" reasoning content for the executor. The lightweight model generates problem-specific reasoning guidance that looks in essence similar to chain-of-thought prefixes.

-  Domain choice doesn't match the experiential memory goal: the choice of the domain (math) is not the best to test experiential memory, which is a far more challenging problem that needs : (i) tasks where past experiences accumulate (ii) scenarios where the agent encounters similar situations repeatedly with an opportunity to handle them better (iii) domains where mistakes made in previous instances inform future behavior etc and (iv) interactive or personalized settings where user history matters (see last suggestion below). Mathematical reasoning problems are largely independent and self-contained. hence, the "experience" being learned here seems more like problem-solving templates rather than experiential memory in the sense the paper claims.

- Baseline selection doesn't validate the core claim: the memory baselines (MEM-0, Dynamic-Cheatsheet, Memento) are not suitable for the task at hand. They were designed for knowledge-intensive tasks, not mathematical problem-solving. This makes it hard to assess whether the improvements you have come from the experiential memory mechanism or "simply" from having an additional reasoning model. THis leads me to my last comment.

- Technically, you are using the "compute power" of two models (the L-PEM and the executor), but then you compare to a "smaller" single model, which is not very fair. The combined parameter coun and inference latency should be compared against single models of similar scale to make a fair efficiency claim.

**Questions:**

- Can you falsify the statement that what you're learning is just the <think></think> reasoning traces for the executor? If not, the framing as "experiential memory" may be overstating what the method actually does.

- I liked the simplicity of the paper, and i found it inspiring. Unfortunately the positioning is weak and there is a mismatch between the experiential goal and what the paper implemented at the end. Moving this work to learning from interactions with a user could be a more valuable contribution.

---

### Official Review · Reviewer_EBZi · 2025-11-01

**Soundness:** 2
**Presentation:** 2
**Contribution:** 2
**Rating:** 4
**Confidence:** 3

**Summary:**

The authors propose L-PEM, a method that embeds experiential memory within the parameters of a lightweight generative model. This approach utilizes a structured memory template for direct generation , enabling a frozen executor LLM to leverage this experience. The primary contributions include:

1. A modular, "plug-in" architecture for parametric experiential memory that operates with a frozen executor, preserving the executor's original capabilities.
2. An experience-generation mechanism that replaces the conventional "store-and-retrieve" RAG paradigm. This aims to overcome the limitations of relevance-based retrieval (e.g., surface-level semantic similarity) by generating task-conditioned, utility-focused guidance.

**Strengths:**

The paper addresses a critical and important research problem: enabling large models to accumulate and reuse prior experiences.

Originality: The approach of optimizing a plug-in experience module via training is somewhat innovative. However, the overall method pipeline itself is common.

Quality & Clarity: The overall method design for the problem is relatively clear and simple, making it easy to understand

**Weaknesses:**

1. Mismatch in Problem Formulation: The paper claims to address the lack of reusing prior experiences. However, the L-PEM model does not seem to memorize or distill experience from the executor's actual past problem-solving rollouts. Instead, it learns to generate plausible-sounding guidance (which could be viewed as "hallucinated memories" or hints) from scratch. This guidance is then optimized via RL based on the executor's downstream task correctness. This process seems to deviate from the core concept of "learning from experience" (which implies accumulating knowledge from specific past events) and aligns more closely with "learning to generate helpful hints."

2. Conflation with Exploration/Hint Generation: This approach appears much closer to research on LLM exploration or generating problem-solving hints, rather than memory. The L-PEM's "experiential memory space" (Problem Analysis, Experience Highlights, Reference Plan)  is functionally equivalent to the structured hints generated by systems designed to guide reasoning (e.g., Google's 'DeepThink' or 'SEED Prover'). The authors should clarify why this is "memory" and not "optimized hint generation," and should ideally compare against such methods.

3. Lack of Experimental Clarity: The experimental section lacks crucial details, making reproducibility and interpretation difficult:
* The "time-to-correct" metric is vaguely defined as "average wall-clock time to obtain a correct answer". This definition is ambiguous. Does this imply re-running the model until a correct answer is produced? If so, what is the sampling strategy (e.g., temperature)? Or is it a calculation like (Per-instance time) / (pass@1 accuracy)? The authors must provide a precise, mathematical definition.
* The RAG baseline setup is underspecified. How was the memory bank for the RAG baselines (MEM-0, DC, Memento)  constructed? Was it built from the same 5k training instances? What were the retrieval parameters (e.g., top-k) used during the pass@1 evaluation? This information is vital for a fair comparison.

**Questions:**

1. Could you please provide a precise, mathematical definition for the "time-to-correct" metric? How is it calculated? Does this metric assume multiple samples per problem? And how does this relate to the selection of the problem set (e.g., was the test set filtered to only include problems the executor can solve with optimal guidance)?

2. The paper studies "experiential memory." A key aspect of memory systems is their ability to improve with more experience (i.e., scaling). The current training uses 5k instances. Could the authors please provide results showing the relationship between the amount of training experience (e.g., 1k, 2k, 5k instances) and the final pass@1 performance? This would be crucial evidence that L-PEM is genuinely "accumulating experience."

3. Could the authors please elaborate on the experimental setup for the RAG baselines ? Specifically, how was their external memory bank constructed, and what retrieval parameters (e.g., top-k) were used during inference for the pass@1 tests?

---

### Note · Authors · 2026-01-06

I have read and agree with the venue's withdrawal policy on behalf of myself and my co-authors.